# Tea-Derived Polyphenols Enhance Drought Resistance of Tea Plants (*Camellia sinensis*) by Alleviating Jasmonate–Isoleucine Pathway and Flavonoid Metabolism Flow

**DOI:** 10.3390/ijms25073817

**Published:** 2024-03-29

**Authors:** Haoming Zuo, Jiahao Chen, Zhidong Lv, Chenyu Shao, Ziqi Chen, Yuebin Zhou, Chengwen Shen

**Affiliations:** 1Key Laboratory of Tea Science of Ministry of Education, Hunan Agricultural University, Changsha 410128, China; zuohaoming@stu.hunau.edu.cn (H.Z.); shaochenyu@stu.hunau.edu.cn (C.S.);; 2National Research Center of Engineering & Technology for Utilization of Functional Ingredients from Botanicals, Collaborative Innovation Center of Utilization of Functional Ingredients from Botanicals and Co-Innovation Center of Education Ministry for Utilization of Botanical Functional Ingredients, Hunan Agricultural University, Changsha 410128, China; 3Key Laboratory for Evaluation and Utilization of Gene Resources of Horticultural Crops, Ministry of Agriculture and Rural Affairs of China, Hunan Agricultural University, Changsha 410128, China

**Keywords:** *Camellia sinensis*, tea polyphenols, drought stress, metabolites, jasmonic acid–isoleucine, flavonoids

## Abstract

Extreme drought weather has occurred frequently in recent years, resulting in serious yield loss in tea plantations. The study of drought in tea plantations is becoming more and more intensive, but there are fewer studies on drought-resistant measures applied in actual production. Therefore, in this study, we investigated the effect of exogenous tea polyphenols on the drought resistance of tea plant by pouring 100 mg·L^−1^ of exogenous tea polyphenols into the root under drought. The exogenous tea polyphenols were able to promote the closure of stomata and reduce water loss from leaves under drought stress. Drought-induced malondialdehyde (MDA) accumulation in tea leaves and roots was also significantly reduced by exogenous tea polyphenols. Combined transcriptomic and metabolomic analyses showed that exogenous tea polyphenols regulated the abnormal responses of photosynthetic and energy metabolism in leaves under drought conditions and alleviated sphingolipid metabolism, arginine metabolism, and glutathione metabolism in the root system, which enhanced the drought resistance of tea seedlings. Exogenous tea polyphenols induced jasmonic acid–isoleucine (JA-ILE) accumulation in the root system, and the jasmonic acid–isoleucine synthetase gene (TEA028623), jasmonic acid ZIM structural domain proteins (JAMs) synthesis genes (novel.22237, TEA001821), and the transcription factor MYC2 (TEA014288, TEA005840) were significantly up-regulated. Meanwhile, the flavonoid metabolic flow was significantly altered in the root; for example, the content of EGCG, ECG, and EGC was significantly increased. Thus, exogenous tea polyphenols enhance the drought resistance of tea plants through multiple pathways.

## 1. Introduction

Environmental stresses cause significant yield losses in agriculture and horticulture worldwide. Abiotic stressors such as light, ultraviolet light, drought, salinity, mechanical damage, etc., directly affect crop yields and horticultural economics [1]. For example, the severe drought in the United States in 2012 led to huge losses in corn and soybean production [2]. In the future, over 44% of the EU’s agricultural imports are projected to be highly susceptible to drought. The severity of droughts is expected to increase by 35% in 2050 compared to the current level of agricultural imports from their respective regions of origin [3]. Increasing reports of drought are also evidence of the dramatic changes in today’s natural environment, which require new solutions to guide agricultural strategies that are resilient to current and future climate conditions.

Drought is one of the most important abiotic stresses worldwide, and excessive drought stress impairs plant growth and reduces yield and quality [4,5]. When drought occurs, plants suffer a variety of damage including membrane lipid peroxidation, protein misfolding, and metabolite destruction [6]. Drought stress leads to stomatal closure and weakened respiratory metabolism, leading to reduced photosynthetic intensity and the accumulation of reactive oxygen species (ROS). Plants have developed a spectrum of physiological, genetic, and metabolic responses to adapt to adverse environments [7]. Furthermore, the pentose phosphate pathway (PPP) serves as a crucial metabolic pathway under environmental stress, compensating for energy and NADH deficiency in the Embden–Meyerhof–Parnas pathway–tricarboxylic acid cycle (EMP-TCA) [4]. Plant-derived hormones also play a pivotal role in plant drought stress by regulating plant growth and a variety of responses [6,8] and regulate these metabolic responses through phenotyping. In addition, stomatal closure-related genes and sugar and phenylpropanoid metabolism in tea plants were regulated by topical application of fulvic acid, 24-epibrassinolide, and zinc to improve drought tolerance [9,10].

Tea plant (*Camellia sinensis* (L.) *O. kuntz*) is an evergreen crop that prefers humid climates. Under field conditions, tea plants are often subjected to drought stress, which has a profound effect on the yield and quality of leaves [6]. With water scarcity, drought has become a major factor affecting field production of the tea plant, resulting in yield reductions of about 14–40% per year in different growing regions [1]. Particularly in regions such as China, India, and Iran, drought plays a greater role in inhibiting the growth of tea plants and adversely affects the management of tea plantations [11,12,13]. The response mechanism of tea plant to drought stress has been reported more, and we need to focus on the drought resistance strategy, which is important for the research and development of drought resistance tools and reagents.

Tea polyphenols have been widely reported as an exogenous stress regulator. EXTP can increase the water absorption capacity and drought resistance of the root, thus enhancing the stress resistance of tea plants [11]. Exogenous catechins directly scavenge ozone into o-quinone in *Zamioculcas zamiifolia* to mitigate ozone damage to *Zamioculcas zamiifolia* [14]. Studies have shown that EXTP can increase the chlorophyll content and photosynthesis rate of the tea plant, which, in turn, promotes tea growth and development. Catechins can also alleviate ozone stress by maintaining chlorophyll content, magnesium content, and stomatal conductance in rice [14]. Drought stress significantly induces the expression of phenylalanine and flavonoid metabolic pathways related to catechin biosynthesis and promotes catechin synthesis in tea plants [15,16,17]. PAL gene expression in tea plant was unaffected by mild drought stress and increased with decreasing salicylic acid content under sustained drought stress [18]. In addition, state-of-the-art histological tools, including metabolomics and transcriptomics, can comprehensively elucidate the changes in photosynthetic system, energy metabolism, and flavonoid pathway gene expression in tea plants under drought stress, demonstrating the response of tea plants to abiotic environments by EXTP [11,19].

Tea plantations pruning leaves and tea leaves with low economic value are potential reservoirs for tea polyphenol extraction. An in-depth understanding of EXTP for the biosynthesis and diversity of specific metabolites and their response pathways in tea plants under drought conditions may provide a novel resistance strategy for the growth and development of tea plants under drought conditions. We hypothesized that the presence of EXTP resulted in improved drought resistance in plants. In this study, the widely cultivated excellent annual variety ‘9808’ was used to jointly analyze how EXTP regulate drought resistance in tea plants using phenotypic changes and metabolomic and transcriptomic data. An in-depth understanding of the effects of EXTP on plant drought resistance will help to provide a theoretical basis for the rational utilization of pruned leaves of tea gardens and tea leaves with low economic value and the development of broad-spectrum plant drought-resistant agents.

## 2. Results

### 2.1. Effects of EXTP on Morphological Characteristics and Physiological Indexes of Tea Plant under Drought Stress

#### 2.1.1. Tea Plant Phenotyping

In this study, two treatment modes, drought and externally applied tea polyphenols, were used for the experiment. Figure 1A shows that EXTP can significantly improve the growth of leaves under drought conditions. Furthermore, EXTP has a significant effect on the growth of the root. In leaves, the TP + DS group maintained the highest tea polyphenol content, with an increase of 66% compared to CK (Figure 1B). Leaf water content was significantly lower in TP, DS + TP, and DS, and lowest in DS, respectively, compared to CK (Figure 1D). The tea leaves in the CK maintained the lowest chlorophyll a (chla) content (Figure 1F). Chlorophyll b (chlb) showed a similar pattern to chla, with a significant increase in chlb content under the DS and TP + DS, and significantly higher than the CK and TP (Figure 1G). EXTP significantly affected the level of carotenoids under drought conditions, with a significant increase in carotenoid content in the DS + TP (Figure 1H). MDA in leaves was significantly higher in DS treatment (Figure 1I). The TP + DS treatment was able to significantly reduce drought-induced MDA accumulation (Figure 1J). This suggests that EXTP pretreatment can alleviate the effect of drought stress on the phenotype of tea plant and can significantly affect root growth.

#### 2.1.2. Anatomical Structure Analysis of Tea Plant Leaves

The stomatal morphology of tea leaves was affected by drought stress and EXTP, and closed stomata were clearly observed under DS, and stomata under TP + DS showed significant opening compared to DS (Figure 2A–D). The highest number of open stomata was observed under TP (Figure 2E). The presence of EXTP resulted in a significant increase in stomatal density (Figure 2F). Compared with the normal group, drought stress significantly reduced stomatal area (Figure 2G). Tea polyphenols led to a significant increase in the number of lateral roots growing in root (Figure 2H). EXTP promoted root elongation (Figure 2I), which was highly significant (*p* < 0.01) compared with both CK and DS. Moreover, EXTP had a strong positive effect on improving root weight (Figure 2J).

### 2.2. Transcriptome Analysis of EXTP Affecting Drought Stress

#### 2.2.1. Transcriptome Quality

For the transcriptome sequencing analysis of the samples consisting of root and leaf (each with three biological replicates), the clean data of each sample reached more than 6 Gb, the error rate (%) was 0.03, and the percentage of Q30 bases was 93% and above (Appendix A).

#### 2.2.2. Differentially Expressed Genes Analysis

In PCA, the first principal component (PC1) explained 39.18% of the variation, and the second principal component (PC2) explained 12.02% of the variation (Appendix A). EXTP had the greatest effect on the transcriptome of roots. Correlation analysis showed that the Pearson correlation coefficients within the eight sample groups were as high as 0.98 or more (Appendix A). The drought conditions showed highly significant differences in gene expression in both leaf and roots, with the presence of EXTP having the greatest effect on gene expression in the root under drought (Appendix A) and less on leaf-related genes, with only 223 genes jointly affected (Appendix A).

#### 2.2.3. Enrichment Pathway Analysis

After obtaining the differentially expressed genes between leaves and roots of tea plants in different treatments, the differential genes were analyzed via KEGG and GO enrichment (Appendix A).

In leaves, the presence of EXTP caused significant enrichment of differentially expressed genes mainly in the antioxidant system as well as in metabolic processes (Figure 3A and Appendix A, Appendix A). KEGG analyses indicated that three metabolic pathways, phenylpropane biosynthesis (ko00940), flavonoid biosynthesis (ko00941), and astragalus formate, diarylheptane, and curcuminoid biosynthesis (ko00945), may be the key pathways by which EXTP affects tea leaves. (Appendix A). In addition, starch and sucrose metabolism (ko00500) may be the EXTP interacting with drought stress in tea leaves, thus affecting the drought resistance of tea plants (Figure 3B).

In the root, EXTP altered the drought resistance of tea plant mainly by affecting the genes related to oxidative metabolism in the root (Figure 3A and Appendix A). In particular, genes related to phenylpropanoid biosynthesis and flavonoid synthesis, as well as two genes regulating ascorbate peroxidase (APX) (TEA019986 and novel.8671), were significantly up-regulated (Appendix A). Flavonoids biosynthesis (ko00941) and phytohormone signaling (ko04075) were significantly different under three conditions, which might be the key pathways of the EXTP affecting the root (Appendix A). Glutathione metabolism and phytohormone signaling pathways, which are closely related to plant drought resistance, were also significantly enriched in the DS + TP-R (Figure 3B).

Comparative KEGG enrichment analyses showed that EXTP significantly affected both phenylpropane biosynthesis (ko00940) and flavonoid biosynthesis (ko00941) in leaves and roots. Moreover, the presence of EXTP affected the antioxidant system as well as multiple metabolic pathways of tea plant, which may be a more important point to improve the drought resistance. The large differences in response patterns between leaves and roots further indicated that EXTP applied to the root under drought stress regulated the resistance of tea plants to drought stress mainly by affecting the differential expression of genes in the root.

### 2.3. Metabolomic Analysis

#### 2.3.1. Metabolomic Data Quality

Appendix A shows the reliable results of QC sample mass spectrometry assay data analysis. Repeatability analyses of metabolites within groups of eight leaf and root samples showed a minimum Pearson correlation coefficient r as high as 0.98 and were significantly different between the four treatment groups for both leaves and roots. (Appendix A). Cluster analysis showed (Appendix A) that the metabolite contents in the four groups of tea plant leaves showed significantly different trends, and the presence of EXTP up-regulated the metabolite abundance of both leaves and roots. The metabolome data were analyzed according to OPLS-DA, the model was validated for each group, and the minimum value of Q2 was 0.904 (Appendix A).

#### 2.3.2. Metabolite Content in Tea Plants

A total of 1055 metabolites were identified from leaves and roots, which were categorized into 11 groups (Figure 4A). Among them, flavonoids (containing 335 metabolites, 32%) were the most abundant, followed by phenolic acids (170 species, 16%) and lipid (98 species, 9%). The levels of alkaloids, nucleotides and derivatives, and amino acids and derivatives were increased in leaves under drought treatment compared to CK, but EXTP reduced them (Figure 4B). However, in the root, the addition of EXTP increased the levels of alkaloids, flavonoids, phenolic acids, and terpenoids compared to DS (Figure 4C and Appendix A). And it is noteworthy that the addition of EXTP under drought conditions caused the levels of amino acids and derivatives to be reduced compared to DS, both in the leaves and in the root. K-MEAN analysis of these 1055 metabolites showed (Appendix A) that they were classified into seven subclasses, with the third class containing the most metabolites (144) and the first class containing only 22 metabolites. The accumulation of these metabolites may be the reason why EXTP affect the drought resistance of tea plants.

The projected importance (VIP) values of ectopic silica in each comparison group were obtained by OPLS-DA (Appendix A). We screened the differential accumulation metabolites (dam) based on VIP > 1 and |FC ≥ 1.5 or ≤0.67. Across all groups of differential metabolites, the EXTP resulted in the up-regulation of many metabolites in the root (Figure 5A), whereas in the leaves, there was a substantial up-regulation of differential metabolites in only the drought condition. We identified the top 10 DAMs that increased or decreased in each comparison group (Figure 5B). In particular, in leaves, the addition of tea polyphenols under drought conditions significantly up-regulated the levels of eupatilin and 1,3-o-dicaffeoylglycerol, and the levels of procyanidin c1 were significantly reduced compared to DS, whereas in roots, EXTP significantly up-regulated the top 10 metabolites in all of them, mainly flavonoids and pheaolic acid, with monogalloyl diglucose and epiafzelechin being the two main up-regulated metabolites. These substances are also more abundant in the tea plant in response to drought stress. Thus, EXTP may alter the drought resistance of the tea plant mainly by affecting the metabolism of substances of flavonoids, tannins, and pheaolic acid.

#### 2.3.3. Effects of EXTP on Metabolites of Tea Plant under Drought Stress

As shown, by CK vs. TP comparison, the differential metabolites reached 39 in leaves, whereas the differential metabolites of EXTP in the root reached 133 (Figure 6A). By DS vs. DS + TP, differential metabolites in leaves reached 29, while differential metabolites of EXTP in the root reached 140 (Figure 6B). EXTP led to the up-regulation of five metabolites in leaves and roots of CK and TP, namely, 1-O-(3,4-dihydroxy-5-methoxy-benzoyl)-glucoside, ethylparaben, benzyl acetate, muconic acid and acacetin-7-o-rutinoside, mainly phenolic acids and flavonoids (Figure 6C). Whereas the co-variable differential metabolites in leaves and roots of DS and DS + TP were neochlorogenic acid, 1,3-dicaffeoylquinic acid, and 15(R)-hydroxylinoleic acid, mainly phenolic acids, and lipids (Figure 6D), we hypothesized that EXTP under drought stress, metabolites connecting leaves and roots may be related to them.

#### 2.3.4. Metabolic Pathway Enrichment Analysis

We analyzed the enrichment of all DAMs in the four treatment groups of CK, DS, TP, and DS + TP using the KEGG database. Appendix A shows the top 20 enriched pathways among the four groups. The addition of EXTP under drought stress resulted in a significant enrichment of pathways including flavonoid biosynthesis (ko00941), flavonoid and flavonol biosynthesis (ko00944) in the root, and a significant enrichment of pathways including arginine and ornithine metabolism (ko00472) in the leaves. This further suggests that EXTP mainly affect the metabolism of flavonoids in the root, thereby altering the drought resistance of tea plant, which is in general agreement with our transcriptional results. Notably, the biological clock (ko04712) of tea plant leaves was significantly affected under drought stress, but the significant effect of drought stress on the biological clock disappeared after addition of EXTP. We hypothesized that the presence of EXTP could improve the drought stress-affected tea plant biorhythms, but the exact mode of effect needs to be further verified.

### 2.4. Integrative Analysis of Transcriptome and Metabolome

Combining transcript analysis and metabolomics studies to understand the response mechanisms of EXTP for drought mitigation under drought stress, DEGs and DAMs of DS and DS + TP were co-enriched in several KEGG pathways (Appendix A). The co-enrichment pathways of genes and metabolites in the leaves of the EXTP-only group mainly included phenylpropanoid metabolism and flavonoid biosynthesis. Interestingly, the co-enrichment pathways of genes and metabolites in tea leaves after the addition of tea polyphenols under drought stress included phenylpropane metabolism, amino acid biosynthesis, arginine and proline metabolism, and glutathione metabolism (Appendix A).

The addition of tea polyphenols led to the co-enrichment of the pathways, including glutathione metabolism, flavonoid biosynthesis, and phenylpropane biosynthesis at gene and metabolite co-enrichment in the root compared to CK, suggesting that they are the core pathways of EXTP in regulating drought resistance in the root. And sphingolipid metabolism was significantly enriched in DS-R_vs_DS + TP-R, suggesting that it may be one of the major response pathways for EXTP to alleviate drought stress (Appendix A).

### 2.5. Analysis of Key Pathways of EXTP in Response to Drought Stress

#### 2.5.1. Analysis of Energy Metabolism in Leaves

Based on the comprehensive analysis of previous findings, the stress regulation of photosynthesis, energy metabolism, and respiratory metabolism in the tea plant under drought conditions is correlated. Therefore, we mapped metabolite content and gene expression levels to relevant metabolic KEGG pathways, including photosynthetic response, starch and sucrose metabolism, glycolysis, TCA cycle, and PPP, starting from the photosynthetic response of leaves (Figure 7).

In the photosynthetic response, the presence of EXTP was screened for genes that differed in photosystem I and photosystem II under drought conditions (Figure 7A). In Figure 7B, we found that EXTP ameliorated the unfavorable effects of drought conditions on photosystems, thereby alleviating drought stress. Subsequently, we explored the Calvin cycle pathway in tea and found that EXTP promoted the increased expression of most genes. Compared with DS, EXTP significantly promoted the expression of PGA, PGAld, and GP final starch formation, and the expression of related key genes was significantly elevated (Appendix A). It indicated that EXTP had a positive mitigating effect on the energy metabolism damage brought by drought stress. However, the expression of SS-related genes was significantly up-regulated under drought conditions during the process of final starch formation, such as TEA009811.

In starch and sucrose metabolism, sucrose content was highest under drought conditions, while EXTP decreased sucrose content in leaves under drought conditions but increased glucose content. Glucose was mainly derived from the decomposition of sucrose in the cytoplasm and starch in the chloroplasts. After drought stress, EXTP elevated the expression of INV and AMY compared to DS. The above results suggest that the co-catabolism of sucrose and starch resulted in an increase in glucose content after the exogenous application of tea polyphenols.

In EMP, HK and PK are considered to be the key rate-limiting enzymes of EMP. The expression of HK- and PK-related genes increased after the application of EXTP, suggesting that the presence of EXTP enhanced the metabolism of EMP under drought stress. The g6p content of glucose was decreased under drought conditions, whereas EXTP after drought caused it to rise. Since g6p is also a feedstock for the PPP pathway, the multiple pathways of respiratory metabolism that consume g6p may have led to a decrease in g6p content. G6PDH and 6PGDH are considered to be the key rate-limiting enzymes in the PPP pathway. After drought stress, the expression levels of G6PDH and 6PGDH genes were significantly elevated in DS, whereas the expression levels of G6PDH and 6PGDH genes were lower in DS + TP than in DS, indicating that PPP was significantly enhanced in DS. DLTA gene expression under drought stress was lower than that under EXTP. Drought treatment resulted in a decrease in the content of most organic acids except citric and succinic acids. The gene expression of most enzymes in TCA was higher than DS after the application of tea polyphenols under drought stress. These results suggest that drought stress resulted in a decrease in TCA intensity, but EXTP activated the genes related to these enzymes possibly through feedback regulation, thus alleviating the low-intensity metabolism brought about by drought stress.

Ru5P is an important component of coenzymes and nucleic acids. However, the content of Ru5P was higher in DS + TP than in DS, suggesting that the presence of EXTP under drought stress may make nucleic acid synthesis and catalytic reactions more stable.

#### 2.5.2. Analysis of Drought Response in the Root

To investigate the effects of EXTP on genes and metabolites related to major drought resistance pathways in tea plant under drought stress, we found that DEGs and GAMs related to sphingolipid metabolism, glutathione metabolism, and arginine metabolism were differentially expressed to varying degrees in the root.

Sphingolipid metabolism is one of the important metabolic pathways in response to drought stress in plants. Under drought stress, the application of EXTP resulted in an increase in the content of phosphoethanolamine (Figure 7J), and the expression levels of the majority of genes were higher than DS, including TEA008834 and TEA033269 in AG, TEA014665 in BG, and TEA010680 in GPP. And EXTP can regulate the drought-induced glutathione metabolic pathway response (Figure 7L). Under the influence of EXTP, the levels of all types of metabolites in the Arginine metabolic pathway showed up-regulation compared to DS (Figure 7K) and were consistent with the transcriptional trends, e.g., novel.21316, TEA019294, TEA016333, and TEA012591. Interestingly, we found that the expression of some key genes related to plant circadian rhythms showed an opposite relationship in the root versus leaves (Figure 7M, Appendix A), such as TEA030941 in PIF3, TEA012558 and TEA012616 in CRY, and TEA033521 in PRR5. The presence of EXTP may regulate the biological rhythms of tea plants, but further validation is needed.

#### 2.5.3. Key Response Pathways Jasmonate Metabolism and Flavonoid Biosynthesis in the Root

JA-Ile was significantly up-regulated in DS-R_vs_DS + TP-R (Figure 8A). Eight genes involved in JA-IL signaling were also regulated by drought stress and EXTP. These include one gene encoding jasmonic acid–amino synthetase (TEA028623), three (TEA00182, novel.22237, TEA032228) genes encoding JAZ containing protein, and four transcription factors MYC 2 (TEA005840, TEA031877, TEA014288, novel.20419).

In total, we screened 37 genes and 23 differential metabolites related to flavonoid synthesis in four groups of treatments in the root (Appendix A). Among the 37 genes, all genes were significantly down-regulated except TEA011908, TEA022042, TEA023790, TEA015423, TEA018051, and TEA013443, which showed up-regulation under drought. EXTP was able to balance drought-induced genetic changes, and most of the genes were significantly up-regulated to promote the expression of flavonoid biosynthetic pathways. Among the 23 metabolites, lignan-7-O-neo orange peel glycoside, catechin gallate, gallocatechin gallate, and theaflavin (TF1) were significantly down-regulated under drought stress, and most of the key metabolites of the flavonoid pathway were up-regulated by the addition of EXTP, which was in agreement with the trend of gene expression (Figure 8B).

## 3. Discussion

Drought has severely tested global agricultural systems in recent years, and as a water deficit stress, drought severely affects plant status and yield. In tea plant, an economic leaf crop, drought stress not only leads to a significant decline in tea quality, but also threatens its survival [20]. Therefore, how to adapt or mitigate drought stress is particularly important in production. Tea polyphenols, an important plant antioxidant, are present in large quantities in tea plants and can be extracted from tea plant residues or prunings as antioxidants but are dependent on the concentration and method of application. In this study, it was found that root application of 100 mg·L^−1^ tea polyphenols could provide better drought resistance to tea plants under drought stress.

### 3.1. ROS Accumulated under Drought Stress Activated the ROS Scavenging System of Tea Plant, but the Addition of EXTP Alleviated ROS Accumulation

The production of ROS, including singlet oxygen, superoxide, hydroxyl radicals, and H_2_O_2_, is a crucial process in all organisms and plays a pivotal role in both abiotic and biotic stresses in plants [21]. Drought stress induces the excessive generation of ROS in tea plants. While ROS can act as signals triggering downstream responses in the plant, their excessive accumulation can lead to cell membrane and protein damage, increased membrane permeability, and ion leakage, resulting in oxidative damage to tea plants [21].

MDA is a suitable marker for oxidative damage and membrane lipid peroxidation, representing the result of membrane lipid peroxidation [22]. This oxidative damage exacerbates cellular dehydration and impairs cellular function [23]. Drought stress increases the MDA content in all groups, indicating that drought stress causes oxidative damage to plant cells. However, EXTP can enhance the levels of antioxidant enzymes and the accumulation of osmotic substances, enhancing the ROS detoxification metabolism in DS + TP. EXTP pretreatment significantly reduces the MDA content in the leaves and roots of tea plants under drought stress, restoring it to normal levels and improving the water relations of tea plants, reducing water loss and absorption under drought stress, potentially increasing water use efficiency.

Notably, we found that EXTP pretreatment of tea seedlings under drought stress promoted significant up-regulation of most genes related to phenylpropane biosynthesis and flavonoid biosynthesis and improved the antioxidant capacity of the root. GO enrichment of differential genes in the root also revealed that GO entries associated with scavenging ROS were significantly clustered under both drought resistance and EXTP pretreatment in the root. This indicates that EXTP pretreatment can enhance the antioxidant capacity of the roots of tea plant under drought stress, alleviating the accumulation of ROS.

### 3.2. Addition of Tea Polyphenols Effectively Improves the Accumulation of Substances Affected by Drought Stress

Drought stress treatments significantly affected the metabolite contents in tea leaves and roots, and the numbers of differential metabolites were all over 100, mainly including amino acids and their derivatives and flavonoids, etc., and they might be substances for tea plants to resist drought stress. The number of differential metabolites in the CK-L vs. TP-L and DS-L vs. DS + TP-L groups was less than half of the number in the drought-induced (CK-L vs. DS-L) group, suggesting that EXTP produced a lesser effect on the leaves of the tea plant, which was consistent with the results of the transcriptome. The number of differential metabolites in the CK-R vs. TP-R and DS-R vs. DS + TP-R groups were all in the range of 130 or more, mainly including flavonoids and phenolic acids, which suggests that EXTP may mainly affect the metabolic processes in the root of *Camellia sinensis* and thus alter the drought resistance of tea plants.

Tea plants can synthesize a diverse range of flavonoids, including anthocyanins, flavanones, flavonols, and flavanols, which are involved in ecological and abiotic stresses of plants [24]. Recently, it has been reported that the tea plant responds to drought stress by affecting the tea plant phenylpropanoid pathway mainly after repeated drought and rehydration [11]. In this study, exogenous additives (polyphenol extracts of tea leaves) mainly affected the accumulation of flavonoids and phenolic acids in the root of tea plants, thereby modulating their drought resistance. Genes of phenylpropane biosynthesis and flavonoid biosynthesis pathways were also significantly enriched.

Alkaloids are a diverse group of secondary metabolites found in plants, and their increased presence has been linked to improved drought resistance [25], with studies showing that infection of *Arabidopsis thaliana* by *Plasmodiophora brassicae* induces the accumulation of feruloyl, coumaroyl, and N-feruloyl [26]. These compounds form polyamine conjugates, serving as natural antioxidants capable of effectively scavenging DPPH, superoxide, and hydroxyl radicals [27]. Furthermore, these alkaloids demonstrated by the up-regulation of genes associated with hydroxycinnamoyl amide synthesis by rhizobacteria promoting plant growth, leading to their accumulation and enhanced antimicrobial capacity [28]. Overexpression of OsNAC6 in rice promotes the synthesis and increased content of niacinamide, reducing drought-induced iron ion levels and preventing hydroxyl radical production, thus enhancing drought resistance [29]. Additionally, under conditions of excess iron, the OsNAC3 gene in rice roots promotes niacinamide synthesis, mitigating tissue damage and cellular metabolic disruption caused by iron toxicity [30]. Sinapine, a natural cationic hydrophilic phenol, selectively lowers mitochondrial oxidative stress levels, effectively entering the mitochondria [31]. In this study, EXTP pretreatment could promote the accumulation of alkaloids such as p-coumaroylputrescine, feruloylputrescine, nicotianamine, feruloylagmatine, and sinapine in the root of tea plant under drought stress, which may help to clear the root of excessive ROS and help to improve the drought resistance of tea seedlings.

### 3.3. EXTP Promote Root Adaptation to Drought Stress in Tea Plant under Drought Stress

#### 3.3.1. EXTP Promote Up-Regulation of Jasmonic Acid–Isoleucine Signaling in Tea Root under Drought Stress

Lipid-derived signaling compounds, jasmonic acid and its derivatives, collectively known as JAs, regulate resistance to abiotic stresses, such as trauma, ultraviolet light, salt, and drought [32,33,34]. The accumulation of higher levels of JAs, its precursor 12-oxo phytodienoic acid, and the active form JA-Iles in root tissues is the main reason for its drought resistance [35]. In the *Arabidopsis* JAO2 mutant, JA-Ile signaling was significantly up-regulated, and metabolites and genes associated with drought resistance were significantly enriched, but ABA was not affected [36]. In rice, exogenous acetic acid treatment of the root rapidly induced jasmonic acid signaling to improve drought resistance but was not dependent on an increase in ABA [37]. In the present study, we found that JA-Ile content in the root decreased under drought stress and increased under drought stress after EXTP pretreatment. JA-Iles gene; two genes, namely, novel.22237 and TEA001821, which regulate the protein synthesis of JAMs; and two genes, namely, TEA014288 and TEA005840, which regulate and transcription factor MYC2, were significantly up-regulated. The content of JA-Ile in leaves tended to increase under both treatments but was not significant. The accumulation patterns of ABA and JA-Ile in the root showed opposite trends and were consistent in the leaves. The accumulation of JA-Ile by ABA in the root showed an antagonistic trend, probably due to the interaction of the JA and ABA signaling pathways at several points [38]. In leaves, the synchronized accumulation of abscisic acid and JA-Ile may be the result of hormonal signaling induced by drought and EXTP. In conclusion, EXTP pretreatment induced the up-regulation of JA-Iles gene expression to promote JA-Ile synthesis, enhanced JA-Ile signaling in the root, and improved drought resistance in tea plants, but the interaction relationship with ABA needs to be further investigated. The significant up-regulation of the flavonoid metabolic pathway may be a consequence of the significant up-regulation of the jasmonate signaling pathway.

#### 3.3.2. EXTP Alter Flavonoid Metabolic Flow in Roots under Drought Stress

Tea polyphenols are the largest group of secondary metabolites in the tea plant, all with phenolic structures, and can be categorized mainly into phenolic acids, brassicas and flavonols, and flavanols and their derivatives. Polyphenols in plants act as non-enzymatic antioxidants capable of scavenging excess ROS to maintain normal cellular metabolic activity [39,40,41,42].

Phenolic acids are an important class of plant secondary metabolites that play key roles in developmental processes. The phenylpropanoid biosynthetic pathway is activated under abiotic stress conditions (drought, heavy metals, salinity, high/low temperatures, and ultraviolet radiation), leading to the accumulation of various phenolic compounds [24]. Phenylpropane biosynthesis may represent an alternative pathway for photochemical energy dissipation with enhanced cellular antioxidant capacity [43]. PAL, C4H, and 4CL are important proteins in the phenylpropane synthesis pathway. In this study, the expression of TEA024587, the gene encoding PAL, TEA016772, the gene encoding C4H, and TEA028833, the gene encoding 4CL, was significantly elevated in drought-stressed tea roots pretreated with EXTP, which promoted a significant increase in the content of phenolic acids, which could help to withstand oxidative stress.

Flavonoids and flavonols have better antioxidant capacity than phenolic acids due to their structural properties. Extracts of flavonoids and flavonols from plants are widely used in medicine and are good in vitro antioxidants [44]. The antioxidant capacity of phloretin [45], eriodictyol [46], dihydroquercetin [47], etc., has been verified. CHS is the first key enzyme in the flavonoid metabolic pathway, and drought stress induced significant up-regulation of the CHS gene in thick-headed amaranth. FLS products of quercetin, myricetin, and kaempferol are the first enzymes of the flavonoid pathway metabolite flow to the flavonoid and flavonol biosynthesis branching pathway, and drought stress induced significant up-regulation of FLS gene in strawberry. In this study, drought stress after EXTP pretreatment induced significant up-regulation of TEA018665, a gene regulating CHS, two genes regulating F3H, TEA034016 and novel.5555, and TEA016601, a gene regulating FLS, in the root, which promoted phloretin, eriodictyol, apigenin-7,4′-dimethyl ether, morin, dihydroquercetin (taxifolin), dihydromyricetin (ampelopsin), isovitexin, and poncirin were synthesized to improve the antioxidant capacity of roots.

Catechins and their derivatives in tea plants are the main products of drought and regulate the plant’s resistance to drought. These substances may also serve as the primary flavonoid antioxidants in tea plants [6]. The content of non-ester catechins (GC, EGC, C, and EC) in Longjing tea leaves decreases with decreasing water availability [48]. Some studies suggest that EGC and EC potentially play a crucial role in tea plants’ resistance to drought. Enzymes such as DFR, LAR, and ANR are essential for directing the flavonoid metabolism towards catechin synthesis. In Longjing tea leaves, under 20% PEG 600-induced drought stress, the expression of genes DFR, LAR, and ANR initially decreases and then increases, promoting an increase in total flavonoid content [48]. In this study, pretreatment with EXTP induced a significant up-regulation of genes TEA031287, TEA027582, novel.12200, novel.21765, and novel.17831, which regulate DFR, LAR, and ANR in the roots of tea plants under drought stress. This led to a significant accumulation of flavonols and their derivatives. It was shown that EXTP could promote flavonoid biosynthetic pathways in roots, especially catechin synthesis.

Glycosylation is a common form of phenylpropanoids present in plants. The specific role of glycosylated phenylpropanoids in plant stress resistance is unclear, but the accumulation of glycosylated phenylpropanoids under stress is a common phenomenon that contributes to stress resistance. UDP flavonoid glycosyltransferase gene expression is up-regulated under drought stress [49,50]. GSA1 encodes a UDP glucosyltransferase that has glucosyltransferase activity for flavonoids and monophenols. Under abiotic stress, GSA1 protects rice from stress by shifting metabolic flux from lignin biosynthesis to flavonoid biosynthesis and the accumulation of flavonoid glycosides that can be readily mobilized [51,52]. In this study, drought stress after EXTP pretreatment induced the up-regulation of three genes encoding UFGT, TEA016972, TEA016979, and TEA031170, in the root, which promoted the accumulation of 56 glycosidic substances, and 37 of them were significantly increased in the root without drought stress by EXTP pretreatment. This suggests that EXTPs can promote the accumulation of polyphenol glycosides in root species of normal-growing tea plants and further increase the polyphenol glycosides after being subjected to drought stress, which improves the drought resistance of tea plants.

In summary, this is an experiment on the drought resistance strategy of tea plants that we made under increasingly severe drought conditions, and the importance of the presence of EXTP for improving drought resistance in the tea plant was demonstrated using multi-omics techniques (Figure 9). Enhancing the synthesis of specific metabolites, such as flavonoids, helps to reduce the oxidative stress state, which leads to a more comfortable life for the tea plant and provides a practical agronomic measure for the conservation strategy of the tea plant under drought conditions. However, our experiment still has some limitations. For example, we failed to further validate the gene network interaction modes affected by EXTP and the difference between the use of EXTP on leaves and root, so in the future, we will further delve into the search for the key gene interaction expression patterns of tea polyphenols affecting drought and further validate their positive effects on drought resistance for this premium agronomic measure for tea plants.

## 4. Materials and Methods

### 4.1. Plant Materials and Sampling

The test material was an annual drought-resistant tea tree seedling ‘9808’propagated from cuttings in Anhua County, Hunan Province (E 113°08′; N 28°18′).

The ‘9808’ tea seedlings, aged one year, were carefully selected for pre-planting. A total of 108 tea plants were transplanted into pots, with three plants and 7 kg of soil per pot, resulting in 36 pots in total. The average height of the tea seedlings was 28 cm. After planting, the potted plants were placed in a transparent greenhouse with a roof and air permeability. The physicochemical properties of the test soil were 0.994 g/kg total nitrogen, 53.8 mg/kg hydrolytic nitrogen, 0.930 mg/kg adequate phosphorus, 52.3 mg/kg fast-acting potassium, 16.8 g/kg organic matter, and pH 4.63. We started the drought treatment on 10 June 2022. The average temperature and humidity of the greenhouse during the treatment period were 31 °C and 80%, respectively. The soil moisture content of the pots was controlled at 25–35% before drought stress, while the control group was watered once every two days to maintain this moisture content. This study simulates the natural drought method. The first group was the control group, after pouring 100 mL pure water in each basin, water was poured every two days (CK); the second group was the drought group, each basin poured 100 mL pure water without watering (DS); the third group was irrigated with 100 mL tea polyphenols (concentration: 100 mg·L^−1^) after drought treatment (DS + TP); and the fourth group was irrigated with 100 mL tea polyphenols (concentration: 100 mg·L^−1^) every two days after pouring water (TP). The soil surface was covered with white gauze to prevent falling leaves from interfering with the concentration of exogenous tea polyphenols (EXTP). Nine days after the treatment, we collected roots [11], one bud, and two leaves and fixed the samples with liquid nitrogen, followed by transcriptomic and metabolomic analyses. The eight groups of samples were labeled as leaf (CK-L) and root (CK-R) in the control group; leaf (DS-L) and root (DS-R) in drought group; and leaf (DS + TP-L) and root (DS + TP-R) in drought treatment group after tea polyphenol pretreatment. Leaves (TP-L) and roots (TP-R) of the normal watering group were treated with tea polyphenols. Each indicator contained three biological replicates. The tea polyphenols were extracted from green tea at a concentration of 98% (tea polyphenols were purchased from Hunan Aijia Biotechnology Co., Ltd., Changsha, China).

### 4.2. Scanning Electron Microscopy

Two × 2 mm tissue samples were carefully excised from clean secondary leaves and promptly placed into centrifuge tubes containing an electron microscope fixative (Servicebio, Wuhan, China). Subsequently, the leaf tissue was vacuum-treated to facilitate sinking and then stored in a 4 °C refrigerator for over 4 h for fixation. The fixed tissue blocks were then removed and immersed in 0.1 mol/L PBS buffer (Servicebio, Wuhan, China) for 15 min each, followed by fixation with 1% osmium acid (Ted Pella Inc., Redding, CA, USA) for 30 min, with three 15-min rinses in between. The tissue blocks were sequentially dehydrated in ethanol solutions of 50%, 70%, 80%, 90%, 95%, 100%, and 100%. Following this, the blocks were dried in isoamyl acetate for 10–20 min. The tissue blocks were affixed to the specimen table with the lower epidermis facing upward and coated with a layer of gold film (60 s) using a vacuum coater (Leica, Wetzlar, Germany). Finally, the processed samples were subjected to observation and image acquisition using a scanning electron microscope (acceleration voltage: 1, 2, 5, 10, 15, and 30 KV, respectively; U8018, HITACHI, Hitachi, Japan). The leaf stomatal length, width, and density were analyzed using Image-Pro Plus 6.0 (Media Cybernetics, Rockville, MD, USA). The calculation formula is as follows [53]:Stomatal density (number·mm^−2^) = stomata number/area.

### 4.3. Malondialdehyde (MDA) Detection

MDA was determined with a plant malondialdehyde kit (TBA colorimetric method). To a total of 0.1 g of leaf and root tissues, we added 1 mL of pre-cooled extraction buffer, homogenized the sample on ice, then centrifuged the sample at 13,000× *g* for 10 min at 4 °C and removed the supernatant for further analysis.

The supernatant was taken for further analysis. We added the assay solution and mixed well, incubated it for 30 min at 95 °C (cover tightly to prevent water loss), cooled it in an ice bath, and centrifuged it at 10,000× *g* for 10 min at 25 °C. We pipetted 200 µL of the supernatant into a 96-well plate and measured the absorbance values at 532 nm and 600 nm using an enzyme counter.

### 4.4. Photosynthetic Pigment Content

The test sample comprised the second leaves, and the leaf tissue was precisely weighed to 0.02 g (with three biological replicates). Subsequently, it was cut into 1 mm-wide strips and placed into a pre-prepared centrifuge tube containing 5 mL of 95% ethanol. The tube was then covered with an opaque plastic bag and stored in a dark location for 12 h, during which it was gently shaken 2–3 times until the leaves turned completely white. The absorbance of the extract solution at 665 nm, 649 nm, and 470 nm was determined using a spectrophotometer (UV-2550, Shimadzu, Kyoto, Japan). The calculation formula is as follows [54]:Chlorophyll a(Chla) content (mg/L) = 13.95 × A665 − 6.88 × A649,
Chlorophyll b(Chlb) content (mg/L) = 24.96 × A649 − 7.32 × A665,
Carotenoid content (mg/L) = (1000 × A470 − 2.05 × Chla − 114.8 × Chlb)/245.

### 4.5. Relative Water Content of the Leaves and Root

The second leaf and root were selected as the test samples, and 0.1 g of tissue was carefully weighed on each side of the central vein of the same leaf (with three biological replicates). One side of the tissue was placed in a centrifuge tube filled with distilled water and left in a dark location for 12 h. Subsequently, the tissue was dried with filter paper and weighed to obtain the saturation weight. The tissue on the other side was subjected to fixation in an oven at 105 °C for 10 min, followed by constant temperature baking at 85 °C until a constant weight was achieved, and then weighed as dry weight. The calculation formula is as follows: leaf and root relative water content (%) = (fresh weight − dry weight)/(saturated weight − dry weight) × 100%.

### 4.6. Determination of the Content of Tea Polyphenols

The polyphenol content was determined using the Folin–Ciocalteu method from the literature. We weighed 0.2 g (accurate to 0.0001 g) of ground tea sample into a 10 mL centrifuge tube and extracted tea polyphenols with 70% methanol on a 70 °C water bath. After centrifugation and volume setting to 10 mL, the test solution (1 mL), Folin–Ciocalteu (5 mL), and 7.5% Na_2_CO_3_ solution (4 mL) were added to the 10 mL centrifuge tubes and left at room temperature for 60 min. The tea polyphenol content was analyzed using a spectrophotometer at 765 nm to determine the tea polyphenol content.

### 4.7. Transcriptome Analysis

The 8 groups of samples of tea leaf and root were used for RNA sequencing (RNA-seq). The RNA integrity and concentration were measured using an Agilent 2100 Bioanalyzer (Agilent Technologies, Inc., Santa Clara, CA, USA). The mRNA was isolated using the NEBNext Poly (A) mRNA Magnetic Isolation Module (NEB, E7490). The cDNA library was constructed using NEBNext Ultra RNA Library Prep Kit for Illumina (NEB, E7530) and NEBNext Multiplex Oligos for Illumina (NEB, E7500) per the manufacturer’s instructions. The constructed cDNA libraries of the leaves and roots were sequenced on a flow cell using an Illumina HiSeq™ sequencing platform. The reference genome can be accessed at http://tpia.teaplant.org/download.html, accessed on 16 December 2022. Transcriptome data results are available for further analysis (Appendix A). DESeq2 v1.22.1 was employed to analyze differentially expressed genes (DEGs), with the Benjamini–Hochberg method used to correct *p*-values. Corrected *p*-values (<0.05) and |log2Fold Change| (≥1) were utilized as thresholds for DEG screening.

### 4.8. Metabolomics Analysis

Metabolite detection and qualitative and quantitative analysis of 8 samples of leaves and roots were undertaken according to the method with three biological replicates [13]. Widely targeted metabolomics analysis was performed by the Met-ware Biotechnology Co., Ltd. (Wuhan, China). The leaves and roots sampled were vacuum freeze-dried in a freeze drier (Scientz-100 F, Zhejiang, Ningbo, China) and then ground into powder using a grinding mill (MM 400, Retsch, Haan, Germany). Next, 100 mg of powder was mixed with 1.2 mL of 70% aqueous methanol (*v*/*v*) and shocked for 30 s six times at a 30 min interval with a Vortex-6 (Kylin-Bell, Jiangsu, China). The homogenate was placed in a refrigerator at 4 °C for 12 h and centrifuged at 12,000 rpm at 4 °C for 10 min. The supernatant was then filtered through a 0.22 µm microporous membrane and stored in a sample bottle.

Ultra-performance liquid chromatography (UPLC) was performed using a Shimadzu Nexera X2 instrument (Shimadzu, Kyoto, Japan) equipped with an Agilent SB-C18 column (1.8 µm, 2.1 mm × 100 mm). The mobile phase A was ultrapure water with 0.1% formic acid, and mobile phase B was acetonitrile with 0.1% formic acid. The gradient procedure was set as follows: 0 min, 5%; 0–9 min, raised to 95%; 9–10 min, 95%; 10–11.10 min, reduced to 5%; and 11.10–14 min, 5%. The column temperature was 40 °C, and the injection volume was 4 µL. Tandem mass spectrometry (MS/MS) was carried out using an Applied Biosystems 4500 QTRAP instrument (ABI, Framingham, MA, USA). Linear ion trap and triple quadrupole scans were obtained using the API 4500 QTRAP UPLC–MS/MS system equipped with an electrospray ionization turbo ion-spray interface. The parameters in the electrospray ionization source operations were set as follows: turbo spray in ion source, and 550 °C for the source temperature; ion spray voltage, 5500 V (positive ion mode)/−4500 V (negative ion mode); ion source gas I, ion source gas II, and curtain gas set to 50, 60, and 25 psi, respectively; and high ionization induction parameters. In addition, 10 μM polypropylene glycol solutions in triple quadrupole mode and 100 μM polypropylene glycol solutions in linear ion trap mode were used for instrument tuning and quality calibration.

Screening was carried out using principal component analysis, orthogonal partial least squares discriminant analysis model, and differential accumulation metabolites (DAMs) with a fold change (FC) of ≥2 or ≤0.5 and a variable importance in projection (VIP) value of ≥1.

### 4.9. Statistical Analyses

Statistical analyses, including the least significant difference (LSD), Student’s *t*-test, and Duncan’s multiple ranges, were conducted using SPSS 25.0 (SPSS Inc., Chicago, IL, USA), with significance indicated by a *p*-value < 0.05. All data are presented as the mean ± standard deviation (SD) of at least three replicates. Principal component analysis (PCA), partial least squares discriminant analysis (PLS-DA), and orthogonal partial least squares discriminant analysis (OPLS-DA) were performed at https://cloud.metware.cn, accessed on 11 January 2023. Graphs were generated using GraphPad Prism v. 8.0.1 (GraphPad Software, La Jolla, CA, USA), Sangerbox (http://vip.sangerbox.com/login.html, accessed on 14 February 2024), and OmicShare Tools (https://www.omicshare.com/tools, accessed on 18 February 2024).

## 5. Conclusions

In this study, we discussed the response mechanism of tea plant growth to drought stress alleviated by EXTP. (1) The presence of EXTP alleviated drought-induced stomatal closure in tea plants, as well as imbalances in photosynthesis and energy metabolism, leading to a significant increase in the abundance of related genes and metabolites; (2) the presence of EXTP reduced the accumulation of ROS caused by drought, lowering MDA content and alleviating oxidative stress through pathways such as glutathione metabolism; (3) the presence of EXTP enhanced the metabolism pathways of JA-Ile and flavonoids under drought conditions, resulting in a significant increase in various stress-resistant metabolites such as theaflavins and catechins, thereby enhancing the drought resistance of tea plants. Therefore, using EXTP to enhance the drought resistance of horticultural plants is feasible, providing new insights for the development of drought resistance strategies in the agricultural field in the future.

## Figures and Tables

**Figure 1 ijms-25-03817-f001:**
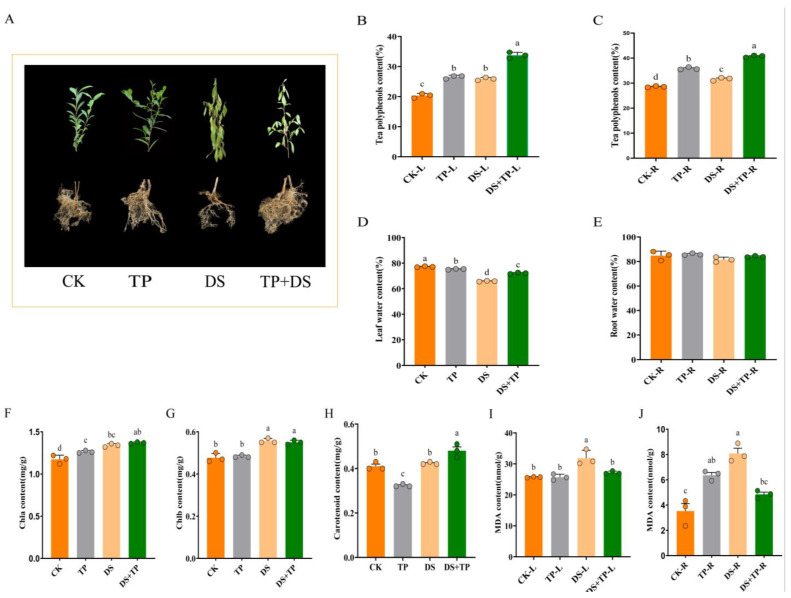
Comparative analysis of (**A**) phenotypes; (**B**,**C**) leaf and root tea polyphenols; (**D**,**E**) leaf and root water content; (**F**–**H**) leaf chla, chlb, and caroteoid content; (**I**,**J**) leaf and root MDA content of tea seedlings under different treatment conditions. Letters a–d indicate significant differences between groups, *p* < 0.05. CK, control group; DS, drought group; TP, normal externally applied tea polyphenol group; DS + TP, externally applied tea polyphenol group after drought.

**Figure 2 ijms-25-03817-f002:**
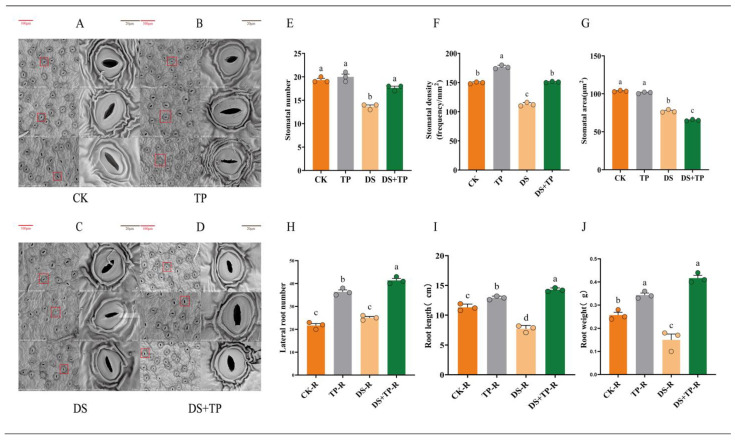
(**A**–**D**) Scanning electron microscope analyses of stomata in CK, TP, DS, and TP + DS leaves; (**E**–**G**) comparisons of stomatal number, stomatal area, and stomatal density in leaves under different treatments; and (**H**–**J**) number of lateral root growth, length of the primary root system, and root mass under different treatments. Red boxes indicate the corresponding positions of the right stomata. Letters a–d indicate significant differences between groups, *p* < 0.05. CK, control group; DS, drought group; TP, normal externally applied tea polyphenol group; DS + TP, externally applied tea polyphenol group after drought.

**Figure 3 ijms-25-03817-f003:**
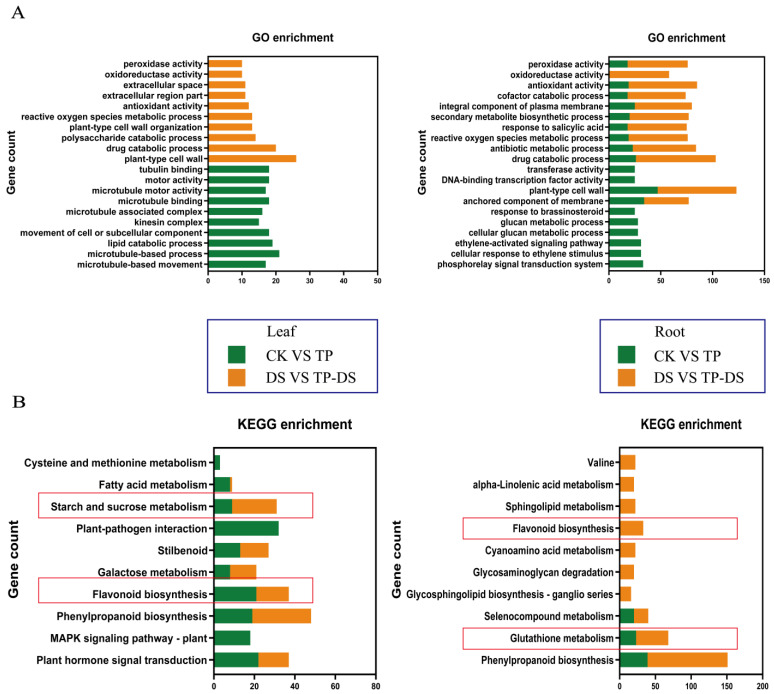
EXTP treatments under normal and drought conditions. Effects on enrichment of tea plant genes GO (**A**) and KEGG (**B**). The presence of EXTP resulted in pathway enrichment of DEGs under different conditions, mainly including the phenylpropanoid pathway, glutathione pathway, flavonoid biosynthesis, etc. Red boxes indicate content that needs attention. CK, control group; DS, drought group; TP, normal externally applied tea polyphenol group; DS + TP, externally applied tea polyphenol group after drought.

**Figure 4 ijms-25-03817-f004:**
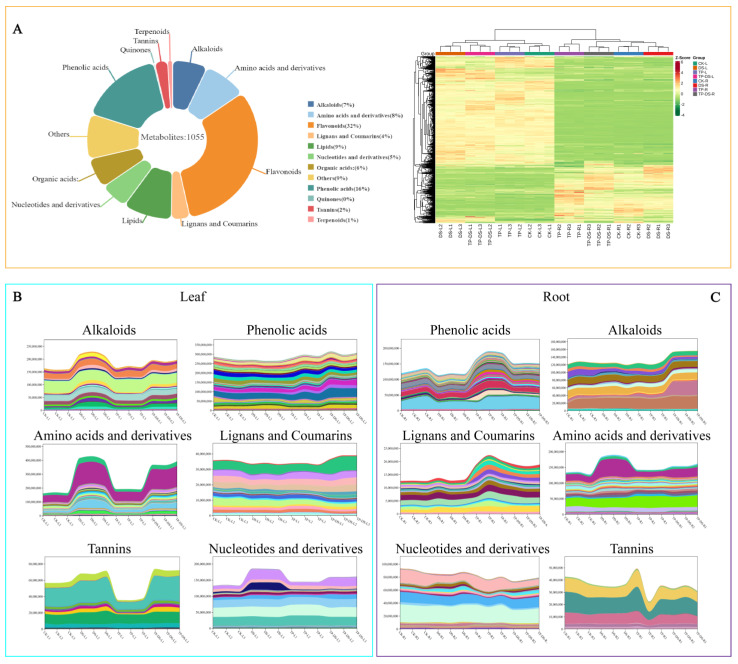
Effect of exogenous tea polyphenols on metabolites of tea plants under drought stress. (**A**) Percentage of total metabolites. River plot of differences in each metabolite taxon in (**B**) leaves and (**C**) roots. Metabolites were classified into 11 categories, with each color representing a compound. The rise and fall of these lines represent changes in metabolite content. Phenolic acids, lignans and coumarins, and flavonoids in the root were significantly increased by EXTP treatment. CK, control group; DS, drought group; TP, normal externally applied tea polyphenol group; DS + TP, externally applied tea polyphenol group after drought.

**Figure 5 ijms-25-03817-f005:**
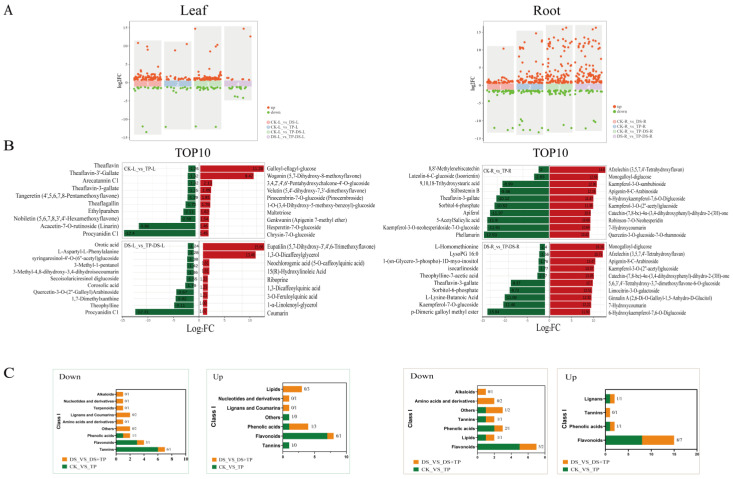
Up- and down-regulation of DAMs. (**A**) Scatter plot showing up- and down-regulated metabolites. (**B**) Up- and down-regulation of the top 10 DAMs. (**C**) Top 10 metabolite taxa to which the DEMs belong. The results showed that EXTP promoted the up-regulation of metabolite levels, especially flavonoids, phenolic acid, and tannins, in the tea root under normal and drought conditions. CK, control group; DS, drought group; TP, normal externally applied tea polyphenol group; DS + TP, externally applied tea polyphenol group after drought; FC, fold change.

**Figure 6 ijms-25-03817-f006:**
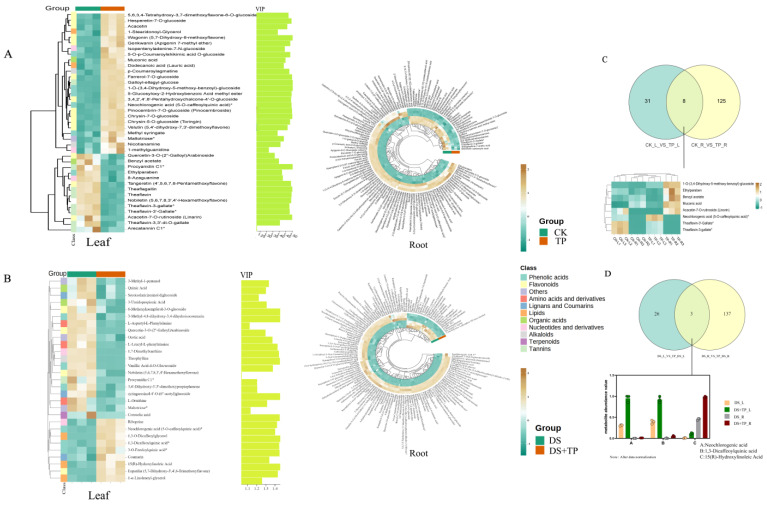
Analysis of EXTP on differential accumulation of metabolites in leaves and roots of tea plant under drought conditions. Heatmap of leaf and root metabolite differences and leaf metabolite VIP values in (**A**) CK and TP, (**B**) DS and DS + TP; Venn diagram of leaf and root differential metabolites and heatmap (or histogram) of co-metabolites in (**C**) CK and TP, (**D**) DS and DS + TP. CK, control group; DS, drought group; TP, normal externally applied tea polyphenol group; DS + TP, externally applied tea polyphenol group after drought.

**Figure 7 ijms-25-03817-f007:**
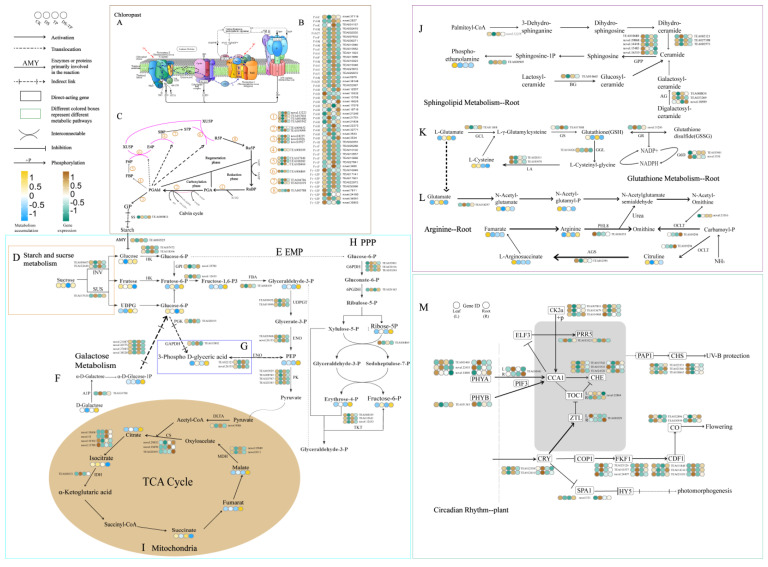
DEGs associated with light reactions (**A**). (**B**) Relative expression levels of DEGs in light reactions. (**C**) Relative expression levels of DEGs in Calvin cycle. (**D**–**I**) Relative expression levels of DEGs in energy metabolism and relative contents of DAMs. (**J**–**L**) Relative expression levels of DEGs in resistance metabolism and relative contents of DAMs. (**M**) Relative expression levels of DEGs in the growth rhythm system. The heatmap colored in blue and yellow indicates metabolite accumulation. The heatmap colored in green and brown indicates gene expression. CK, control group; DS, drought group; TP, normal externally applied tea polyphenol group; DS + TP, externally applied tea polyphenol group after drought.

**Figure 8 ijms-25-03817-f008:**
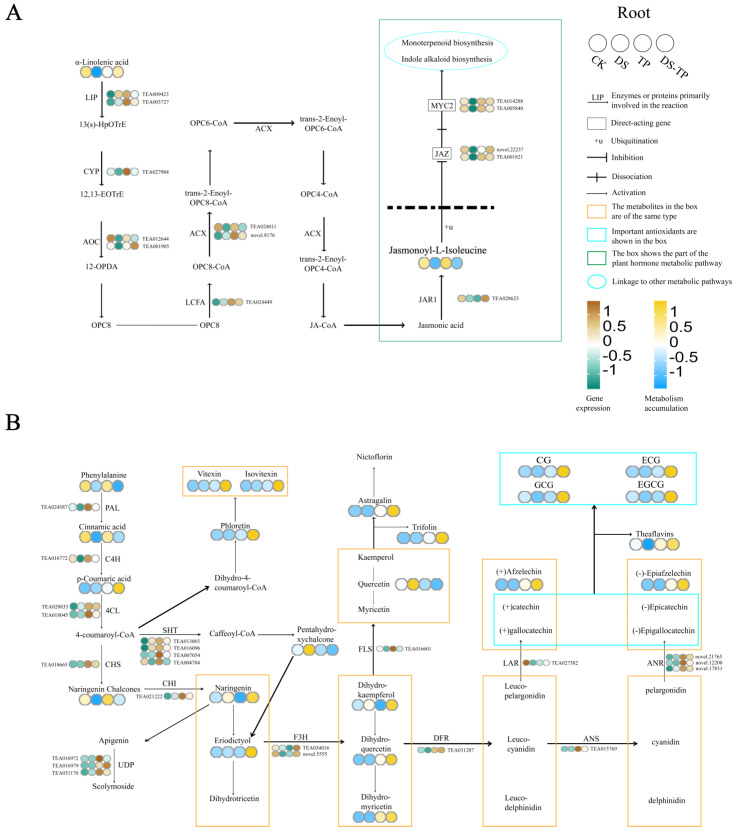
Analysis of the coexpression of (**A**) jasmonate metabolism and (**B**) flavonoid biosynthesis in roots. The heatmap colored in blue and yellow indicates metabolite accumulation. The heatmap colored in green and brown indicates gene expression. CK, control group; DS, drought group; TP, normal externally applied tea polyphenol group; DS + TP, externally applied tea polyphenol group after drought.

**Figure 9 ijms-25-03817-f009:**
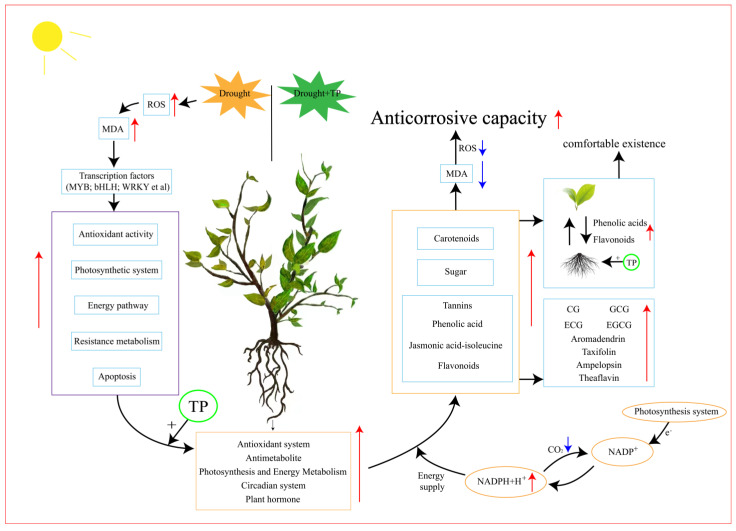
The putative steps for EXTP to mitigate the growth response of tea plants under drought stress. Blue boxes represent corresponding substances or transcription factors, purple boxes represent metabolic pathways that do not positively affect tea plants, orange boxes represent co-responsive substances or metabolic pathways that actively resist stress in tea plants, orange human circles represent photosynthetic system responses, black arrows indicate signaling or substance effects, red arrows indicate up-regulation of corresponding gene expression or metabolite abundance, and blue arrows indicate down-regulation of substances.

## Data Availability

Data available on request from the authors.

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
