# Peer review of "Tea-Derived Polyphenols Enhance Drought Resistance of Tea Plants (*Camellia sinensis*) by Alleviating Jasmonate–Isoleucine Pathway and Flavonoid Metabolism Flow"

_ijms, 2024, doi:10.3390/ijms25073817_

Round 1
Reviewer 1 Report
Comments and Suggestions for Authors
The manuscript describes the investigation of the effect of exogenous tea polyphenols on the drought resistance of tea plant by pouring 100 mg-L-1 of exogenous tea polyphenols into the root under drought. The exogenous tea polyphenols were able to promote the closure of stomata and reduce water loss from leaves under drought stress, and the mechanism was explored.
The topic is of practical significance, the scope is adequate and the methods are appropriate.
Minor point that affects reproducibility is that the SEM operating parameters have not been introduced.
Although the Abbreviations have been listed at the end of the manuscript, at first appearance of each what it stands for should also be given for immediate explanation. It appears what GO stands for has not been explained.
Result presentation and interpretation are of high quality. The usage of varying types of diagram is good.
Moderate English editing is needed.
Hence, can be accepted for publication after minor revision, addressing the following additional points:
Fig.2 SEM micrographs do not have scale bars.
Fig. 1, 2 the bar charts, the usage of three data points for each bar is fine. However, in Fig. 1B, H, J, and others, some data points are directly beneath others (same x, different y) instead of being next to one another (different x, different y). Is it supposed to be like this?
Line 722-724, suggest adding ref [42-44, 44b] 44b= Systematic characterisation of the structure and radical scavenging potency of Pu'Er tea () polyphenol theaflavin, 2019

minor editing
Reviewer 2 Report
Comments and Suggestions for Authors
The authors address the very topical issue of drought and its effect on tea tree quality. They use tea polyphenols as "anti-stress" substances. The introduction is somewhat lacking in detail on the effect of drought on tea plants. The authors mention the influence of phytohormones in the defence against drought, but only in general terms, without being specific. The results are based on graphs, which, although they complement the text, are often small and make the descriptions of the axes harder to read. I recommend that the graphs be modified. The text of the results is somewhat unbalanced, as in some sections the authors give the values obtained, whereas in other sections they only give the results without specifying the experimental data. I recommend unification. The discussion is rather descriptive in places. I have more reservations about the methodological part, where it is stated that the tea plants were grown in 7 litre containers, but the characterisation of the growing medium, i.e. its physico-chemical properties, is completely lacking. Furthermore, the text lacks growing conditions, e.g. temperature and light regime, humidity ... It might be useful to include the GPS coordinates of the growing areas. The authors also assessed the water content of the leaves, but it is not specifically stated whether this is RWC or water saturation deficit.
Reviewer 3 Report
Comments and Suggestions for Authors
I really enjoyed this manuscript. I have some comments:
Title:
- 'Exogenous' stands for external application or coming from other origin? After reading, you can understand, but I think should be clearer in the title
- 'Ppathway' by 'Pathway'
Abstract:
- Not sure about the use of the word 'popular' is very scientific terminology
Introduction:
- I would contextualize the drought effects on tea plantations (different locations or countries), as the general overview is well known
- I'll focus this introduction in the aspects you work in the manuscript, namely metagenomics, genetic pathways involved...
- I feel the references could be more and better contextualized
- Zamioculcas zamiifolia (or thereafter in 'Discussion', Camelia sinensis, line 622), scientific name in italics
- Some abbreviations are not first-appearance full explained (as EXTP) but later, other not even full described (as PAL). Despite some are well-known, this is necessary to avoid confusions and enhance reproducibility
Results
- Take care with some terms (as 'Phnolic acid', line 285; or 'alkaoids', line 297). Take a look, because I detected some like this
- Really requires more extended description of methods and tools working in this manuscript...
- In general, this section is very dense, you may consider alleviating the load and resume main points
Discussion
- Some concepts have been 'discussed' in 'Results' section. In order to not feel it repetitive, please ensure the concepts are not duplicated in the manuscript
- Not sure how you arrive to 100 mg/L as the concentration to test. Seems very high, so not sure if easy to extract enough (and with enough quality, as they degrade easily) from debris
- Too many introduction-like explanations in this section. On the other hand, many approaches included here seems like a duplication of the results... Which are already largely exposed before, don't feel is necessary to do it again here
- too dense for the content, to be honest
Materials and Methods
- Not a good description on how the EXTP have been obtained (???)
- If the tea selected variety is drought-resistant, are not the results less signification?
- The use of 'CK' as control or mock is not adequate, as could be confused with cytokinins abbreviation
- One bud and 2 leaves are representative enough? Doubt it...
- The description of some methodologies are excessively vague (as in 'Determination of the content of tea polyphenols' or 'transcriptome analysis', or 'metabolomic analysis'). Really need to provide with details in order to guarantee the quality assessment and the reproducibility of the work
General:
- No pretty sure of the premises: EXTP is not referenced how to provide, but moreover, the proposal of using EXTP to use in tea plants is kind of weird, as they used to be dedicated to tea consuption, so now a part of the production is dedictated to mitigate drought? Is as well a lost in the production for consuption... My meaning is that I understand and value the idea, but the description of the approache and the further use of this approach is very vague and may consititute of a self.contradictory case of study
Comments on the Quality of English Language
Just some minor word-misspelling
Round 2
Reviewer 3 Report
Comments and Suggestions for Authors
Thanks for considering my comments, I think now is more readable and this make the work more impactful. I would recommend adding a brief explanation about the origin and testing conditions for the EXTP as you did in the review comments, but in your main text. This would be helpful to clarify and avoid confusions just maybe add that this is a preliminary approach that may need optimization, but it's already showing promising results. I know you included some, but the selection of the concentration still seems not clear. I hope your work is the beginning of more interesting stories about tea genetic regulation under drought and possible biotech applications. Once again, I enjoyed reading and the topic seems interesting to me. Thanks for understanding my concerning and hope this is useful.